# Farmers’ Willingness to Gather Homesteads and the Influencing Factors—An Empirical Study of Different Geomorphic Areas in Chongqing

**DOI:** 10.3390/ijerph19095252

**Published:** 2022-04-26

**Authors:** Yan Yan, Qingyuan Yang, Kangchuan Su, Guohua Bi, Yuanqing Li

**Affiliations:** 1School of Geographical Sciences, Southwest University, Chongqing 400715, China; yanyan0806@email.swu.edu.cn (Y.Y.); biguohua@email.swu.edu.cn (G.B.); xslyq@swu.edu.cn (Y.L.); 2Western Rural Sustainable Development Laboratory, Chongqing 400715, China; 3School of National Governance, Southwest University, Chongqing 400715, China; sukangchuan@swu.edu.cn

**Keywords:** homestead agglomeration, farmers’ willingness, binary logistic regression model, influence factor, Chongqing City

## Abstract

Research purpose: to analyze farmers’ willingness to gather homestead and its influencing factors, so as to provide decision-making basis for the rational layout of rural homestead. Methods: questionnaire, logistic model. The results are as follows. (1) Farmers’ willingness to gather homesteads is highest in dam areas, followed by hilly areas, and is lowest in mountainous areas. (2) The respondents’ age, family support ratio, housing structure, whether the access road is paved, and the distance from the main road have significant negative impacts on farmers’ willingness to gather homesteads, while homesteads being idle, the service life of the house, the type of daily energy use, and whether they are far from relatives after relocation have significant positive impacts on farmers’ willingness to gather homesteads. (3) The main influencing factors of farmers’ homestead agglomeration in dam areas are the idle situation of a homestead, housing structure, the service life of the house, and whether they are satisfied with their current homestead residence. (4) The main influencing factors of farmers’ homestead agglomeration in hilly areas are the age of the respondents, the proportion of family workers, and whether they accept the relocation and are far from relatives. (5) The main influencing factors of farmers’ homestead agglomeration in mountainous areas are the age of the respondents, the ratio of family support, the housing structure, and whether the access road is paved. We conclude that there are significant differences in farmers’ willingness to gather homesteads and the influencing factors in different geomorphic areas. Policy makers should formulate differentiated homestead agglomeration optimization schemes and design the optimization paths of homestead agglomeration on the basis of geomorphic classification and subregion.

## 1. Introduction

Moderate rural homestead gathering is an element of rural spatial reconstruction and an important means of optimizing the spatial patterns of urban and rural land. With rapid worldwide urbanization, the decline of the countryside has become a global problem [1], and large numbers of rural people have moved to urban areas for work and resettlement, resulting in great changes in the relationship between people and land in rural areas. Many homesteads are idle or inefficiently used. Some village configurations have become hollow [2,3,4]. In addition, this situation has caused the inefficient use of a large number of public service facilities. At present, the spatial layout of rural settlements in the hilly areas of southwest China is formed mostly by independent choice and lacks unified planning, resulting in fragmented and disorganized layouts [5], The layout of settlements is influenced by topography, geological conditions, and traffic. Homesteads show a piecewise or group distribution, and their development momentum is greatly constrained [6]. The 19th session of the National Congress of the Communist Party of China proposed the implementation of a rural revitalization strategy with the general aims of “prosperous industry, pleasant environment, civilized countryside, effective governance and wealthy living”, marking a new stage of rural development in China. In May 2019, the Central Committee of the Communist Party of China and the State Council issued the Opinions on Establishing a Sound Institutional Mechanism and Policy System for Integrated Development of Urban and Rural Areas, which proposed the following: “In line with the planning, use control and respect for farmers’ wishes, allowing county governments to optimize the layout of village land. Effectively use the rural scattered stock of construction land. Promote the development of unified homestead area standards around the provinces. Award those who gather incremental homesteads. Compensate those who withdraw from stock homesteads.” The introduction of policies such as rural revitalization and new urbanization will certainly trigger the spatial gathering of agricultural production factors, particularly farmers and homesteads [7]. Local governments are guiding scattered villagers and new farming households to concentrate village layouts, improve living environments, and complement infrastructure and public service facilities. In view of the current scattered situation of homesteads in southwestern hilly and mountainous areas, it is necessary to implement suitable gathering policies in the process of village planning and new rural construction to improve the efficiency of rural land use and reduce repetitive and inefficient state investment in infrastructure and public service facilities. Therefore, in the stage of comprehensive implementation of the rural revitalization strategy, optimizing the spatial development pattern of the countryside, improving the link between people and land, and improving rural habitats to moderately concentrate and optimize rural homesteads are urgent needs. However, to meet the rural revitalization and rural construction requirements, plans for moderately gathering rural homesteads should fully respect farmers’ wishes and analyze the main factors affecting their willingness to participate in centralized living in order to optimize the spaces for rural production, living, and ecology, and improve the rural living environment.

At present, countries use space planning as a means of space governance. From the perspective of respecting residents’ wishes, formal institutions are established in Europe to collect residents’ opinions and views [8]. Coastal countries propose that territorial spatial planning should be implemented at the municipal level, and communities should participate in the planning [9]. In the process of implementing territorial space planning, the UK pays special attention to infrastructure construction to benefit residents [10]. Germany adopts a hierarchical structure to formulate spatial planning; each level is supported by corresponding laws and regulations, and emphasizes public participation [11]. In China, in May 2019, the “Several Opinions of the CPC Central Committee and The State Council on establishing and Supervising the Implementation of the Territorial Space Planning System” further clarified that territorial space planning consists of four major systems, namely, the approval system for planning compilation, the implementation supervision system, the policy and regulation system, and the technical standard system, as well as “five levels and three types”. It is divided into three categories: national, provincial, city, county, township, and general plan, detailed plan, and related special plan. Territorial spatial planning is also composed of “five levels and three categories”, that is, five levels of national, provincial, city, county, township, and three categories of overall planning, detailed planning, and related special planning. At the same time, in July 2021, the newly revised “Regulations on the Implementation of the Land Administration Law “was released, which further refines the requirements of the new “Land Administration Law” on establishing a land and space planning system, and it clarifies the specific path for the compilation and implementation of land and space planning. Trans-regional administrative planning and rural planning should be emphasized in the formulation of spatial planning, which is the guide for national spatial development [12]. The government should coordinate farmers’ production, living, and ecological space, and guide the development boundary of rural settlements [13]. However, China has a special land system, and the right to use rural homestead belongs to farmers. Therefore, the decision-making willingness of farmers on homestead must be emphasized in the implementation of spatial planning. Few studies have focused directly on the causes, characteristics, and evolutionary patterns of homestead gathering, but many studies have investigated the morphological and typological distribution of rural gathering [14,15,16,17,18]. The spatial gathering pattern of rural settlements is aggregative, random, and dispersive as a result of the combined influence of industrialization, counterurbanization, globalization, tourism, and emerging media [19,20,21,22]. Due to the outflow of rural population and other reasons, the concentration points show the development modes of extinction, expansion, filling, and integration [23,24,25,26]. Additionally, researchers have found that moderate homestead gathering is conducive to the layout of public service facilities [27,28] and the improvement of farmers’ quality of life [29,30]. It is also an effective way to reconfigure rural settlement space and improve the human living environment [31,32,33]. Moderate homestead gathering is an effective way to optimize the stock of homesteads and an important means of rural structural adjustment [34,35,36], and it can also drive systemic urban development [37]. The focus should be on environmental satisfaction, social networks, and suitable farming distances [38,39,40]. Some scholars have also proposed promoting the community management model to improve the efficiency of rural homestead utilization [41] or improving the efficiency of use through internal remediation of homesteads [42,43,44]. However, homestead gathering optimization is a systemic project that is influenced not only by natural factors, economic factors, and policies, but also by many other aspects, such as ideology, ethics, religious beliefs, agricultural production practices, and culture [45,46,47]. Some scholars believe that the study of the evolution of the characteristics of the rural homestead land portfolio and its relationship with the livelihoods of farmers has led to important breakthroughs in guiding the planning and management of rural settlement sites [48]. Farmers are the real users of land in rural settlements, meaning they have a great influence on changes in land use at the level of thought and behavior [18]. Their livelihood decisions have a strong influence on the structure of land use within rural settlements [49,50,51]. Therefore, a full study of the factors influencing farmers’ decisions is an important part of the implementation of homestead gathering optimization. Studies have focused mainly on farmers’ willingness to withdraw from homestead and farming land [52,53,54], and studies on farmers’ willingness to gather homesteads have covered only economically developed areas and plains farming areas [55,56]. Research on farmers’ willingness to gather homesteads in a southwestern hilly mountainous region is lacking. Due to the significance of economic and social development in the hilly mountainous regions of southwest China, it is necessary to improve the living environment, supporting infrastructure, and services, and properly reconstruct rural spaces in the hilly mountainous regions in the process of modernization. These issues should also be the focus of current research in Chinese human geography.

Based on the above analysis, in the era of fully implementing the new rural revitalization strategy, we adopted the participatory rural evaluation method, obtained a questionnaire survey from farmers, conducted an in-depth analysis of farmers’ willingness to agglomerate homesteads and the influencing factors in different geomorphic areas of the mountainous and hilly areas of southwest China, and explored the path of appropriate agglomeration of rural homesteads. The findings of this study will be helpful in guiding the full implementation of rural revitalization and building a beautiful countryside.

## 2. Materials and Methods

### 2.1. Study Area

Chongqing is a province-level municipality located in southwestern China in the upper reaches of the Yangtze River. It is located between 105°11′~110°11′ E and 28°10′~32°13′ N in the center of the transitional area between the Qinghai–Tibet Plateau and the middle and lower reaches of the Yangtze River Plain. The south and north ends of the city are at high elevations, and the Yangtze River valley in the middle zone is low. The topography is undulating. The Daba Mountains, Wushan Mountains, Qiyao Mountains, and Wuling Mountains are located east and southeast of the city. The western and eastern parts of the city are dominated by hills, low mountains, and other landform types. To comprehensively understand the willingness of Chongqing farmers to gather homesteads and the influencing factors, we selected sample sites for field research based on regional development differences, topographical differences, and locational differences. In selecting the sample points, we considered two perspectives. First, according to the development pattern of Chongqing, the Banan and Tongnan Districts of the metropolitan area, Liangping District of the northeastern Chongqing city group, and Shizhu and Fengdu Counties of the southeastern Chongqing city group were selected as the study areas. Second, township-level research samples were selected according to different landform types. There are eight major landform categories in Chongqing: medium mountain, low mountain, high hills, medium hills, low hills, gentle hills, mesa, and platform. In this paper, the landforms are divided into three categories: mountain region (medium mountains and low mountains), hill region (high hills, medium hills, and low hills), and platform region (platforms and flat dams). The mountainous region is represented by Zhongyi Township of Shizhu County and Sanjian Township of Fengdu County. The hilly area is represented by Shitan Town in Banan District and Zhushan Town in Liangping District. The platform area is represented by Tai’an Township in Tongnan District (Figure 1). The characteristics of the study area are shown in Table 1.

### 2.2. Data Sources and Processing

(1)Data for the homestead polygon area and the corresponding number of farming households come from five sources: district and county planning departments, the preliminary data of the Third National Land Survey provided by the Bureau of Natural Resources, the rural homestead polygon data of DLTB, high-resolution remote sensing images, and field research. Based on these sources, the number of farm households corresponding to the homestead polygon was obtained, and the spatial clustering characteristics of the homesteads were analyzed.(2)Socioeconomic and economic data come from discussions and exchanges with township officials, township and village statistics for past years, township and village planning documents, relevant vector data, and relocation and ecological migration statistics. Based on these data, the research team was able to analyze the socioeconomic development of homestead gathering.(3)Data reflecting farmers’ willingness to gather homesteads were obtained through a questionnaire survey of farmers. The team selected five townships in five districts and counties and surveyed 10 to 15 randomly selected farm households in each village of the selected townships from early April 2021 to early July 2021. The team used a participatory survey and assessment method to conduct one-on-one interviews with the farmers and paraphrase questionnaire items to ensure that the farmers understood them. The team visited 40 villages and distributed 500 questionnaires. After invalid questionnaires were eliminated, the final number of valid questionnaires was 482. The effectiveness of the questionnaires was 96.4%. The characteristics of the farm household sample are shown in Table 2. The percentage of interviewees older than 50 years was 78.21%. The percentage of those whose education level was primary school or who had not attended school was 71.26%. The data indicate that the majority of people living in rural areas are elderly and have a low education level.

### 2.3. Methods

#### 2.3.1. Model Selection

Logistic analysis is widely used in regression models where the dependent variable is a dichotomous variable. Farmers’ willingness to live centrally is a dichotomous variable dominated by categorical data. Therefore, the team used the logistic regression method with dichotomous dependent variables to establish a model of farmers’ willingness to gather homesteads, applied the maximum likelihood estimate to solve the regression parameters, and conducted an overall test based on probability values. P denotes the probability of the event occurring, and (1 − P) denotes the probability of the event not occurring. The dependent variable indicator of this study is the willingness of farm households to participate in centralized housing. If they are willing to gather, the dependent variable value is set to 1. If they are unwilling, the dependent variable is set to 0. Constructing a logistic model between the dependent and independent variables, and setting the independent variables to *i*, we obtain $b_{0}+b_{1}x_{1}+b_{2}x_{2}+\cdots +b_{i}x_{i}=\sum_{i=1}^{n}b_{i}x_{i}$, where the constant terms *b*_0_, *x*_0_ are 1. Then, the logistic probability function can be expressed as Equation (1):(1)$$\mathrm{prob}\left(\mathrm{type}\right)=f\sum_{i=1}^{n}b_{i}x_{i}=\frac{1}{1+\exp\left[-\sum_{i=1}^{n}b_{i}x_{i}\right]}$$

Suppose $\sum_{i=1}^{n}b_{i}x_{i}=Z$; we multiply the numerator and denominator by exp(z) to obtain Equation (2):(2)$$\mathrm{prob}\left(\mathrm{type}\right)=\frac{1}{1+\exp\left[-z\right]}=\frac{\exp\left[z\right]}{1+\exp\left[z\right]}$$

Prob(type) indicates the probability that farmers are willing to choose homestead agglomeration. According to the classification method in this paper, farmers’ homestead agglomeration intention is studied on the basis of different geomorphic types. Therefore, the model is modified to form four logistic probability regression models. Farmers in the whole region are represented by w; farmers in flat dam areas are represented by d; farmers in hilly areas are represented by h; and farmers in mountainous areas are represented by m, specifically expressed as follows:(3)$$\mathrm{prob}\left(w\right)=\frac{\exp\left[z_{w}\right]}{1+\exp\left[z_{w}\right]};\;z_{w}=\sum_{i=1}^{28}b_{i}x_{i}=b_{0}+b_{1}x_{1}+b_{2}x_{2}+\cdots +b_{28}x_{28}$$
(4)$$\mathrm{prob}\left(d\right)=\frac{\exp\left[z_{d}\right]}{1+\exp\left[z_{d}\right]};\;z_{d}=\sum_{i=1}^{28}{b_{i}}^{\prime }x_{i}={b_{0}}^{\prime }+{b_{1}}^{\prime }x_{1}+{b_{2}}^{\prime }x_{2}+\cdots +{b_{28}}^{\prime }x_{28}$$
(5)$$\mathrm{prob}\left(h\right)=\frac{\exp\left[z_{h}\right]}{1+\exp\left[z_{h}\right]};\;z_{h}=\sum_{i=1}^{28}{b_{i}}^{\prime \prime }x_{i}={b_{0}}^{\prime \prime }+{b_{1}}^{\prime \prime }x_{1}+{b_{2}}^{\prime \prime }x_{2}+\cdots +{b_{28}}^{\prime \prime }x_{28}$$
(6)$$\mathrm{prob}\left(m\right)=\frac{\exp\left[z_{m}\right]}{1+\exp\left[z_{m}\right]};\;z_{m}=\sum_{i=1}^{28}{b_{i}}^{\prime \prime \prime }x_{i}={b_{0}}^{\prime \prime \prime }+{b_{1}}^{\prime \prime \prime }x_{1}+{b_{2}}^{\prime \prime \prime }x_{2}+\cdots +{b_{28}}^{\prime \prime \prime }x_{28}$$

#### 2.3.2. Variable Setting

The centralization of farm households must be based on the voluntary participation of farmers. In essence, homestead gathering is their choice. With reference to the research results of Chinese and foreign scholars and the availability of data, we selected the following specific variables. (1) Personal characteristics: Generally, there are large individual differences in farmers’ age, gender, and education level, and their lifestyles vary. (2) Household characteristics: Farmers have different family structures, nonfarm income ratios, and numbers of elderly individuals and students in their homes, forming differentiated household characteristics. Thus, their willingness to choose homestead agglomeration also differs. (3) Housing situation: There are large differences in the area, location, and size of homesteads owned by different farmers, as well as the ages of houses and other main structures. The degree of utilization varies from normal utilization to partial idleness, seasonal idleness, perennial idleness, and other types. Regarding the age and structure of houses, the longer a house has been used, the lower its structural stability is likely to be. All-wood or stone-and-wood structures have certain safety risk factors that may affect the owners’ willingness to relocate and cluster. (4) Infrastructure situation: Traffic and road conditions in different villages vary widely. Villages with better socioeconomic development have better road facilities, and farmers in those areas are more satisfied with their current living environment. Villages with poorer economic conditions have poor infrastructure conditions, making farmers’ traffic and living conditions inconvenient. Thus, farmers in these villages may be more willing to concentrate their homesteads. (5) Social interaction and living conditions: Social interaction is considered in terms of whether farmers prefer quietness or liveliness, the distance between them and their relatives, and whether they have a harmonious relationship with the surrounding neighbors. (6) Individual subjective cognitive factors: It has been shown that residential satisfaction plays an important role in farmers’ attachment to places [38]. Therefore, we considered whether they are satisfied with the current homestead living environment and the current homestead gathering scale. (7) Current conditions of the village area: Natural geographical conditions and economic and social conditions are influencing factors of the spatial accumulation of land by rural residents [20]. In summary, we set the willingness of farmers to gather homesteads as the dependent variable and analyzed specific factors affecting this willingness in different geomorphological regions by using 28 indicators in seven aspects as independent variables: personal characteristics, household characteristics, housing situation, infrastructure situation, social interaction and living situation, individual subjective cognitive factors, and current conditions of the village. The statistical description of each variable is shown in Table 3.

## 3. Results

Among the households in the survey sample, 230 were willing to gather, accounting for 47.7% of the total number of farmers surveyed, and 252 were not willing to gather, accounting for 52.3%. Farmers’ homestead agglomeration intention varied in different landscape regions. The ratio of farmers who agreed with homestead gathering was 48.56% in the mountain region, 58.70% in the hill region, and 62.5% in the platform region. Therefore, we focused on farmers’ homestead gathering intentions and the factors influencing them in different geomorphological regions.

First, SPSS 25.0 (IBM, Chicago, IL, USA) software was used to diagnose collinearity for data from 482 farmers in different zones and topographical conditions in Chongqing to rule out the possibility of severe multicollinearity in the samples. “Whether the home has electricity” x16 was excluded. The results showed that all 482 households in the sample had electricity, meaning this factor was not meaningful to the calculation. Second, multivariate regression of willingness to gather was performed on the sample using the same software, and marginal utility was further estimated to form better curve-fitting results, as shown in Table 4. According to the statistical principle, if the coefficient estimates are positive, there is a positive relationship between the dependent variable and the independent variable, provided that the model fitness is good, and the coefficient estimates pass the significance test. If the converse is true, there is an inverse relationship. Accordingly, the factors influencing the willingness of farmers to relocate to centralized housing in the whole region, platform region, hill region, and mountain region are explained.

### 3.1. Analysis of Factors Influencing Farmers’ Willingness to Gather Homesteads in the Whole Area

The regression results show that at the 5% significance level, *x*_2_ has a significant negative effect on the dependent variable prob(w). This indicates that the older the farmer is, the fewer opportunities there are for off-farm employment, the more dependent he or she is on familiar surroundings, and the less willing he or she is to live centrally. *x*_6_ has a significant negative effect on the dependent variable prob(w). The dependency ratio is the ratio of the number of elderly people and students in the household to the total number of people in the household. The results show that the more people a household has to provide for, the lower its willingness to cluster. On the one hand, seniors are reluctant to relocate. On the other hand, the larger the ratio, the greater the financial burden on the family. According to Maslow’s demand theory, farmers in this situation pursue more material needs and have not yet considered the demand for quality of life, so their willingness to gather is low.

*x*_9_ and *x*_11_ have significant positive effects on the dependent variable prob(w). The high degree of homestead idleness indicates that the main members of these farming households have moved to cities to work or buy houses, do not rely on agricultural production activities as their main source of income, and are less dependent on rural land resources. Therefore, their willingness to cluster is stronger. The longer the houses have been used, the stronger the farmers’ willingness to renovate them, and the more willing they are to enjoy the policy benefits associated with homestead gathering. Therefore, their willingness to gather is stronger. *x*_10_ has a significant negative effect on the dependent variable prob(w). The better the homestead structure is, the more financial resources the farmers have spent renovating their houses, and these farmers are not willing to move again. Therefore, their willingness to gather is low.

*x*_13_ has a significant negative effect on the dependent variable prob(w). Among farmers whose access roads were not paved, 76.52% were willing to gather. Poor road conditions seriously hinder their productivity and reduce their quality of life. At the same time, their agricultural products cannot be marketed and distributed in a timely manner. These farmers prefer to improve their living conditions through homestead gathering; thus, their willingness to gather is stronger. *x*_17_ has a significant positive effect on the dependent variable prob(w). The proportion of natural gas use in rural Chongqing is only 4%, and the proportion of gas tank use is approximately 30%. Farmers who use gas tanks or natural gas have certain demands for their living conditions and want to have more comfortable and convenient living conditions through homestead gathering. Those who use more primitive energy sources are comfortable with the status quo and are not willing to change, so their willingness to gather is lower.

Regarding social interaction and individual subjective cognitive aspects, *x*_20_ has a significant negative effect on the dependent variable. The farther a farmer is from the main road, the stronger his or her willingness to gather homesteads will be. The main reason is similar to that for the availability of paved roads. *x*_25_ has a significant positive effect on the dependent variable. Farmers who do not accept relocation farther away from their surrounding neighbors or relatives have a lower willingness to gather.

### 3.2. Analysis of Factors Influencing the Willingness of Farmers to Gather Homesteads in the Platform Region

At the 5% confidence level, x_9_ has a significant positive effect on the dependent variable prob(d), and *x*_10_ and *x*_12_ have a significant negative effect on the dependent variable prob(d). These findings indicate that the higher the degree of homestead idleness is, the stronger the willingness of farmers to gather homesteads. The better the structure of homesteads and when they have business situations, such as rentals, the more farmers tend not to cluster. *x*_23_ has a significant negative effect on the dependent variable prob(d). That is, the more dissatisfied farmers are with their current homestead living conditions, the more willing they are to improve their living conditions through gathering policies.

### 3.3. Analysis of Factors Influencing Farmers’ Willingness to Gather Homesteads in the Hill Area

At the 5% confidence level, *x*_2_ has a significant negative effect on the dependent variable prob(h). This indicates that the older hill area farmers are, the more reluctant they are to relocate. *x*_7_ has a significant positive effect on the dependent variable prob(h). This indicates that the higher the number of household workers in the hill region, the higher the farmers’ willingness to gather. A higher number of household workers indicates a higher nonfarm income. This group of farmers will respond positively to the homestead agglomeration guidance policy. *x*_25_ has a significant positive effect on the dependent variable prob(h). These farmers have a herd mentality, and their decisions will be influenced by those of their neighbors.

### 3.4. Analysis of Factors Influencing Farmers’ Willingness to Gather Homesteads in the Mountain Region

At the 5% significance level, *x*_2_, *x*_6_, and *x*_10_ have significant negative effects on the dependent variable prob(m). The research found that in remote Shizhu County, there are more all-wood or stone-and-wood housing structures, accounting for 53.96% of the houses. These housing types have certain safety problems; thus, farmers in these areas will be more willing to gather. A total of 76.52% of farmers whose homesteads were not connected to paved roads were willing to gather.

## 4. Discussion

### 4.1. Application and Refinement of the Model

Based on the research data and analysis of the study area, we modified a model to study farmers’ willingness to gather homesteads from the perspective of different landscape types. When dividing the landscape types to select research sites, we first considered the overall development pattern of Chongqing and selected sample sites from the metropolitan area, northeastern Chongqing city group, and southeastern Chongqing city group. We portrayed the willingness of farmers in different landscape regions with socioeconomic development and other factors as the premise, found that there were obvious regional differences in farmers’ willingness to gather, and analyzed the underlying reasons. The findings are more refined than those in some studies that also used logistic models, which in turn enriches the research methodology for farmers’ decisions regarding homestead utilization [20,54]. In addition, the selection of indicators took into account not only the real situation of individuals and families and their thinking about the future but also the current conditions of the village area. We mainly considered the market development, social services development, and cooperative establishment of the village. These conditions play an important role in the decision making of farmers in the study area.

### 4.2. Extension of Study Results

The development of rural settlements has a spatial pattern of natural growth under the influence of many factors and lacks effective and reasonable guidance. In the context of the comprehensive implementation of rural revitalization, the future demand for rural public facilities should be fully considered, the willingness of farmers to gather homesteads should be fully understood, the influencing factors should be analyzed, village planning should be taken as the guide, and rural homestead gathering should be guided with regional differences in mind. The homestead layout should be comprehensively optimized and adjusted. The rural contracted land and homesteads should be reformed to increase farmers’ land property income and thus provide reference samples for the optimization of rural settlement layout. In order to encourage farmers to live together in the future, it is suggested to start from the following aspects. First, the government should provide some financial support, design attractive residential areas, and establish a strong organizational structure to guarantee and supervise the development of this series of work. Second, reasonable relocation compensation mechanism should be established to protect the practical interests of farmers. Third, real estate property rights certificates should be issued for relocated households to eliminate farmers’ concerns. Fourth, the government should establish community self-governance organizations and community management in residential areas to ensure effective management and autonomy of residential areas.

### 4.3. Future Homestead Gathering Optimization Path

In the context of the new era, the main contradictions of our society have changed. The need for a good life has replaced the need for a material culture. Comprehensive improvement of rural habitats is the requirement of the present day. The proportion of individual polygons with fewer than five households was high in different geomorphological zones, but there were significant differences. The mountain area accounted for 83.95%, the hill area accounted for 82.12%, and the platform area accounted for 70.06%. This shows that the layout of homestead in Chongqing is decentralized, which is not conducive to the matching of infrastructure and public service facilities, thus affecting the comprehensive improvement effect of human settlements. Therefore, it is necessary to fully start the pilot work of moderate homestead gathering in southwestern hilly and mountainous areas, according to the topographical and geomorphological conditions and socioeconomic conditions of this region. To conduct “multi-planning and integration” practical village planning, according to the topographical and geomorphological conditions and socioeconomic conditions of the southwestern hilly and mountain areas, we need to start with a cluster of 10 or more households in mountains and hills and gather homesteads within areas with convenient transportation and accessible production facilities. In the platform area, the government should take a 20-family cluster as the starting point and gradually guide scattered homesteads to gather while taking into account local architectural and cultural characteristics and residents’ living habits. Additionally, the planning and design requirements for homestead gathering should aim to improve the quality of the rural living environment and promote “beautiful home” construction in the southwestern hilly and mountain areas.

### 4.4. Limitations

In this paper, farming distance was set as a variable that was not significantly correlated with the dependent variable as measured by the econometric model. This is probably because residents are accustomed to the existing distance [40]. However, the field research found that the main factor affecting farmers’ willingness to gather homesteads in the hill area was the problem of farming distance after relocation. Therefore, future studies should consider adding the variable “expectation of farming distance after homestead gathering” and collect relevant data. Farmers who cannot transfer their land in the hill area are more dependent on the land. The main reason for their reluctance to relocate to centralized housing is the distance to their farms. When homestead gathering planning is carried out in the future, reforms should be made to set up professional cooperatives or introduce enterprises to realize rural production centralization and agricultural large-scale operation. This approach can optimize the ecological space of rural production and life through a reasonable, scientifically designed homestead layout.

## 5. Conclusions

Based on the characteristics of the rural human–land system in southwestern hilly and mountainous areas, especially the excessively scattered and disorganized homestead spatial characteristics and the level of socioeconomic development, this study considers village planning according to local conditions, guides moderate homestead gathering and the centralization of the rural population, and provides guidance for reform to achieve the integration and coordination of economic, social, and ecological benefits. By analyzing the spatial characteristics of rural homesteads and the willingness of farmers in the southwestern hilly and mountainous areas to gather, the following conclusions were obtained:(1)The spatial layout of rural homesteads in the southwestern hilly and mountainous areas is generally scattered and messy. The proportion of scattering varies in different geomorphic regions. The characteristic is most obvious in the mountain area, followed by the hill area and platform area. At the same time, the willingness of farmers to gather homesteads varies in different landscape types due to geographical location and economic conditions. Farmers’ willingness to gather homesteads is highest in the platform region, followed by the hill region, and is lowest in the mountain region. This pattern is consistent with the current spatial gathering characteristics of homesteads, indicating that the use of homesteads by farmers directly affects the spatial form and layout of rural village settlements. Future optimization of homestead spatial layout should fully consider and respect farmers’ choices and wishes and develop differentiated homestead gathering optimization schemes by region.(2)Additional factors influence the willingness of farmers to gather homesteads. In the hill and mountain regions, which have many natural environmental constraints and relatively low levels of economic and social development, government policy guidance and policy encouragement are key to the planning of moderate rural homestead gathering. From a general point of view, personal characteristics, family characteristics, housing utilization, infrastructure, social interaction and living conditions, and individual subjective perceptions are factors that influence farmers’ willingness to gather homesteads. Farmers are very different individually and have different needs. With the premise of safeguarding their fundamental interests, the government should actively intervene in village planning to guide the rational layout of rural residential land, complement infrastructure and public service facilities, and continuously improve the human living environment.(3)From the perspective of different geomorphic terrain areas, the main factors affecting farmers’ willingness to gather homesteads in the platform area include both the utilization of homesteads and satisfaction with the living environment. The infrastructure of the platform area and the economic status of the villages are better, and the utilization of homesteads by farmers reflects their dependence on homesteads for their survival and their operation of homesteads. These circumstances directly affect farmers’ homestead agglomeration decisions. The main factors influencing the willingness of farming households in hilly areas to gather homesteads are the age of the respondents for individual characteristics, the ratio of the number of laborers for family characteristics, and the subjective cognitive factors of whether to accept distance from relatives after relocation. The four main factors affecting the willingness of farm households in mountainous regions to gather homesteads are individual characteristics, family situation, house utilization, and individual perceptions. New agricultural subjects should be introduced to actively promote the reform of rural contracted land in hill areas, change the way of life and production of farmers in the hilly and mountainous areas of southwest China, and support the comprehensive implementation of the rural revitalization strategy.

## Figures and Tables

**Figure 1 ijerph-19-05252-f001:**
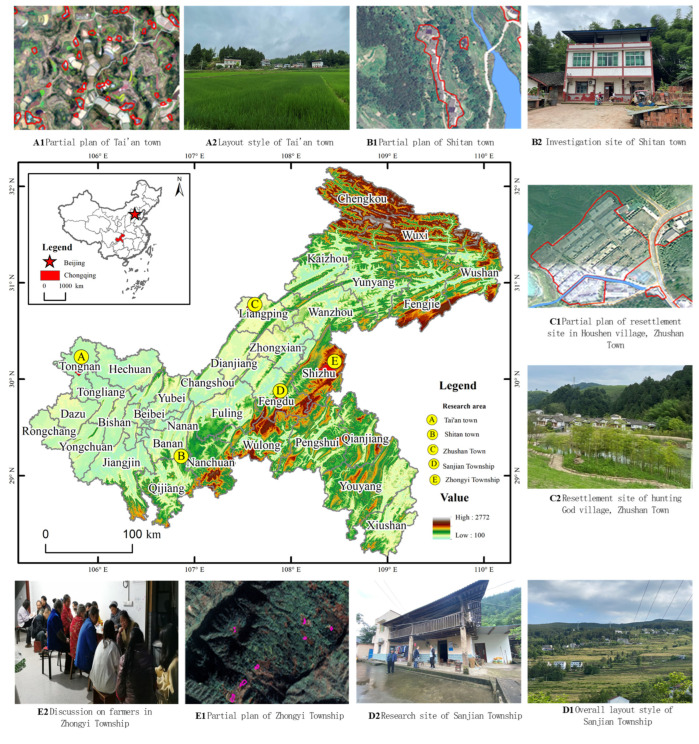
Survey area overview.

**Table 1 ijerph-19-05252-t001:** Basic characteristics of the study area.

Details	Platform Area	Hill Area		Mountain Area	
	Tai’an	Zhushan	Shitan	Sanjian	Zhongyi
The measure of area (km^2^)	60.81	48.73	52.09	62.70	160
Population density (person/km^2^)	605.11	163.76	299.44	220.99	51.56
Location	It is located in the southwest of Tongnan District.	It is located in the west of Liangping District.	Located in the south of Banan District.	Located in the southwest of Fengdu County.	It is located in the middle of Shizhu County.
Topographic conditions	The altitude is between 163 and 346 m, the overall terrain is high in the north and south, low in the middle, the slope is between 0 and 53°, and the overall terrain is flat.	The altitude is between 421 and 1047 m, which is the landform of “two mountains with one trough”.	The altitude is between 520 and 1132 m. The terrain belongs to low mountain and hilly landform. The terrain is high from north to south and low from east to west.	The altitude is 236~1200 m, showing the trend of “three mountains and two rivers”.	The altitude is between 777 and 1892 m, and the slope is between 0 and 67°. The whole township is dominated by steep slopes and less flat terrain.

**Table 2 ijerph-19-05252-t002:** Characteristic distribution of farmer samples.

Indicators	Category	Number	Rate (%)
Interviewee Age (years)	Under 29	12	2.49
	30~39	16	3.32
	40~49	77	15.98
	50~59	161	33.40
	60 and above	216	44.81
Interviewee Education Level	Illiteracy	132	27.30
	Never Went to School	212	43.96
	Primary School	111	22.99
	High School	22	4.60
	Junior College and above	6	1.15
Number of Interviewed Family Members (persons)	1~3	227	47.13
	4~6	236	48.85
	7 and above	19	4.02
Ratio of Actual Number of Laborers in the Surveyed Households (%)	0~30	83	17.24
	30~70	177	36.78
	70~100	222	45.98

**Table 3 ijerph-19-05252-t003:** Index system and statistical description.

Variable Type	Variable	Code	Variable Description	Variable Type	Average	Standard Deviation
Personal Characteristics	Respondent Gender	*x* _1_	Male = 1; Female = 0	Dummy variables	0.68	0.469
	Respondent Age	*x* _2_	Unit: Years	Field observation	59.44	13.42
	Respondent Education Level	*x* _3_	Primary school and below = 1; Middle school = 2; High school = 3; Junior college and above = 4	Dummy variables	1.36	0.63
Household Characteristics	Total Household Income	*x* _4_	Unit: CNY 10,000	Field observation	5.43	5.77
	Percentage of Nonfarm Income	*x* _5_	Unit: %	Field observation	0.53	0.82
	Family-Dependent Population Ratio	*x* _6_	Ratio of number of elderly people and students in the household to the total number of people in the household, unit: %	Field observation	0.48	0.36
	Household Labor Ratio	*x* _7_	Unit: %	Field observation	0.32	0.30
Housing Situation	Area of Family Homestead	*x* _8_	Unit: m^2^	Field observation	114.22	38.29
	Idleness of Homestead	*x* _9_	Using = 1; Partially idle = 2; Seasonally idle = 3; Perennially idle = 4	Dummy variables	1.14	1.03
	Housing Structure	*x* _10_	Concrete structures = 3; Brick-wood or stone-wood = 2; Soil-wood or all wood = 1	Dummy variables	2.19	0.93
	Housing Structure	*x* _11_	≤10(year) = 3; 10~20(year) = 2; 20~30(year) = 1; ≥30(year) = 0	Dummy variables	1.46	1.24
	Housing Rental Status (true or false)	*x* _12_	Operating or rental situation = 1; No operating or rental situation = 0	Dummy variables	0.11	0.31
Infrastructure Situation	Whether Road to House Is Paved	*x* _13_	True = 1; False = 0	Dummy variables	0.90	0.30
	Availability of Streetlights	*x* _14_	True = 1; False = 0	Dummy variables	0.51	0.50
	Whether House Has Centralized Water Supply	*x* _15_	True = 1; False = 0	Dummy variables	0.92	0.26
	Whether Home Is Electrified	*x* _16_	True = 1; False = 0	Dummy variables	1.00	0.00
	Types of Daily Energy Use	*x* _17_	Natural gas = 3; Gas tank and electricity = 2; Wood, coal and electricity = 1	Dummy variables	1.38	0.56
Social Interaction and Living Conditions	Whether Live Near Relatives	*x* _18_	True = 1; False = 0	Dummy variables	0.41	0.49
	Relationship with Neighbors	*x* _19_	Good = 3; Normal = 2; Bad = 1	Dummy variables	2.78	0.45
	Distance from Main Road	*x* _20_	≤200 = 3; 200~1000 = 2; 1000~3000 = 1; ≥3000 = 0; Unit: m	Dummy variables	2.07	0.97
	Distance from Town	*x* _21_	≤3000 = 3; 3000~6000 = 2; ≥6000 = 1; Unit: m	Dummy variables	2.43	0.77
	Distance from Other Farms	*x* _22_	≤500 = 3; 500~3000 = 2; ≥3000 = 1; Unit: m	Dummy variables	2.53	0.61
Individual Subjective Cognitive Factors	Satisfaction with Current Homestead Living Conditions	*x* _23_	Satisfied = 3; Normal = 2; Unsatisfied = 1	Dummy variables	2.65	0.70
	Satisfaction with Current Gathering Scale of Homestead	*x* _24_	Satisfied = 1; Unsatisfied = 0	Dummy variables	2.70	0.63
	Accept Distance from Relatives after Relocation	*x* _25_	True = 1; False = 0	Dummy variables	1.34	0.72
Current Conditions of the Village	Village Economic Status	*x* _26_	Good = 3; Normal = 2; Bad = 1	Dummy variables	2.45	0.76
	General Infrastructure Status of Village	*x* _27_	Good = 3; Normal = 2; Bad = 1	Dummy variables	2.51	0.64
	Topography of Village	*x* _28_	Platform = 3; Hill = 2; Mountain = 1	Dummy variables	1.26	0.61

**Table 4 ijerph-19-05252-t004:** Regression results of logistic model parameters of farmers’ homestead agglomeration in different regions.

Variable Type	Code	Whole Area (prob(w))	Platform Area (prob(d))	Hill Area (prob(h))	Mountain Area (prob(m))
Personal Characteristics	*x* _1_	0.114 (0.517)	−0.106 (0.101)	−0.646 (0.892)	0.190 (0.340)
	*x* _2_	−0.010 * (0.024)	−0.755 (0.039)	−0.009 * (0.033)	−0.026 * (0.016)
	*x* _3_	0.282 (0.687)	0.736 (0.763)	0.195 (0.611)	0.168 (0.286)
Household Characteristics	*x* _4_	−0.211 (0.045)	−0.192 (0.076)	−0.363 (0.074)	0.478 (0.033)
	*x* _5_	−0.454 (0.276)	0.303 (0.193)	0.977 (0.186)	−0.944 (0.482)
	*x* _6_	0.000 * (0.968)	−0.281 (0.084)	−0.906 (1.300)	−0.040 * (0.599)
	*x* _7_	−0.868 (0.797)	0.149 (0.347)	0.004 * (0.535)	0.107 (0.657)
Housing Situation	*x* _8_	−0.508 (0.006)	−0.101 (0.020)	0.411 (0.014)	−0.107 (0.004)
	*x* _9_	0.001 * (0.789)	0.029 * (0.228)	0.658 (0.663)	−0.879 (0.226)
	*x* _10_	−0.019 * (0.735)	−0.023 * (0.134)	−0.199 (0.880)	−0.006 * (0.270)
	*x* _11_	0.003 * (0.829)	−0.353 (0.034)	−0.623 (0.312)	0.649 (0.195)
	*x* _12_	−0.603 (0.826)	0.010 * (0.033)	−0.621 (1.297)	−0.069 (0.548)
Infrastructure Situation	*x* _13_	−0.030 * (−0.720)	−0.274 (0.039)	0.102 (0.957)	0.013 * (0.653)
	*x* _14_	0.259 (0.506)	−0.456 (0.094)	−0.158 (0.771)	−0.403 (0.349)
	*x* _15_	0.305 (0.874)	0.323 (0.088)	−0.716 (0.182)	−0.103 (1.496)
	*x* _17_	0.010 * (0.328)	−0.353 (0.558)	−0.676 (0.639)	−0.250 (0.376)
	*x* _18_	−0.066 (0.489)	0.080 (0.127)	0.556 (0.969)	0.388 (0.311)
Social Interaction and Living Conditions	*x* _19_	−0.167 (0.827)	−0.090 (0.304)	−0.218 (0.710)	−0.537 (0.354)
	*x* _20_	−0.044 * (0.018)	0.300 (0.614)	0.944 (0.672)	−0.017 * (0.204)
	*x* _21_	0.326 (0.632)	0.100 (0.582)	0.329 (0.953)	0.174 (0.247)
	*x* _22_	−0.278 (0.865)	0.099 (0.023)	0.428 (0.628)	−0.947 (0.239)
Individual Subjective Cognitive Factors	*x* _23_	−0.490 (0.766)	−0.009 * (0.392)	0.139 (0.191)	0.515 (0.276)
	*x* _24_	0.286 (0.015)	−0.998 (0.092)	0.054 (0.721)	−0.239 (0.303)
	*x* _25_	0.001 * (0.645)	0.036 (0.277)	0.000 * (0.467)	0.000 * (0.218)
Current Conditions of Village Area	*x* _26_	−0.995 (0.220)	0.519 (0.705)	0.986 (0.000)	−0.285 (0.364)
	*x* _27_	0.449 (0.258)	0.751 (0.356)	0.867 (0.028)	0.159 (0.516)
	*x* _28_	−0.283 (0.604)	−0.190 (0.026)	0.327 (0.962)	0.330 (2.358)

Note: * indicates significance at the 5% level. The values in parentheses are the standard errors of the coefficient estimates.

## Data Availability

Not applicable.

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
