# Peer review of "Farmers’ Willingness to Gather Homesteads and the Influencing Factors—An Empirical Study of Different Geomorphic Areas in Chongqing"

_ijerph, 2022, doi:10.3390/ijerph19095252_

Round 1
Reviewer 1 Report
"Farmers' willingness to gather homesteads and the influencing factors—An empirical study of different geomorphic areas in Chongqing "
The literature review is too focused on China. There is no other than the Chinese perspective on the topic in question. Especially the issues of creating different homesteads systems in rural areas in different conditions should be the subject of the review. What is the role of spatial planning tools today? What are the ways (at the state policy level) to interfere with the issues under study, in different conditions of different countries?
Chapter 2 - The content of lines 182-185 is a typical part of the discussion
After '2.2 Data sources' there follows the chapter '2.3 Data analysis', and between them there is a table with the characteristics of the area. The nomenclature of these chapters indicates some additional research, and this is just a description of the research area and can be included as a 'Study Area' without separating sub-chapters.
The authors divided the article into:
"Study area, data source and analysis" (which does not contain any analyzes)
Then the following chapter was placed: "Variables and models" (where there is actually a part of the methodology and the results).
The next chapter is called "5.Discussion"
The structure of the article requires a deep reorganization and ordering. It seems necessary to separate the sections "Study Area, Methodology, Results and Discussion.
A large part of the discussion is of the nature of wishes, and not very specific:
"We recommend that moderate gathering be encouraged ..."
“Homesteads should be gathered in areas with convenient transportation…”
“The scattered and fragmented layout of homesteads should be strictly controlled“
Changing people's residence (especially related to moving them to another place) or encouraging a specific behavior when building houses (or generally - choosing a place of residence) are difficult issues to implement in practice. What are the specific authors' suggestions? Creating local zooning plans? Financial incentives? Changes in residence in the prescriptive mode resulting from the legal provisions enacted for this purpose? What time perspective are we talking about?
There is no description (in the introduction) of the spatial planning system in China (concerning rural areas). What tools does it have at the level of legal regulations regulating such processes? To what extent do residents participate in this? To what extent is it possible to take their wishes into account?
Line 404. "The main reason for their reluctance to relocate for centralized housing is the distance to their farms"
Can these actions not be correlated with China's widespread land consolidation program ???
"Advantages of Willingness Optimization Model" Has the research really formulated the concept of such a model and proved its validity?
Reviewer 2 Report
In the Abstract not seen research aim, just object.
Why farmers (as a specific group) should show their willingness to gather homesteads?
Why it is important for them and different areas?
Round 2
Reviewer 1 Report
I looked carefully at the changes made by the authors. In my opinion, the present form of the manuscript is of good quality and may be published.